# Health-Related Quality of Life and Experiences of Sarcoma Patients during the COVID-19 Pandemic

**DOI:** 10.3390/cancers12082288

**Published:** 2020-08-14

**Authors:** Eugenie Younger, Alannah Smrke, Emma Lidington, Sheima Farag, Katrina Ingley, Neha Chopra, Alessandra Maleddu, Yolanda Augustin, Eve Merry, Roger Wilson, Charlotte Benson, Aisha Miah, Shane Zaidi, Anne McTiernan, Sandra J. Strauss, Palma Dileo, Spyridon Gennatas, Olga Husson, Robin L. Jones

**Affiliations:** 1The Royal Marsden NHS Foundation Trust, London SW3 6JJ, UK; eugenie.younger@rmh.nhs.uk (E.Y.); alannah.smrke@rmh.nhs.uk (A.S.); emma.lidington@rmh.nhs.uk (E.L.); sheima.farag@rmh.nhs.uk (S.F.); yolanda.augustin@rmh.nhs.uk (Y.A.); eve.merry@nhs.net (E.M.); charlotte.benson@rmh.nhs.uk (C.B.); aisha.miah@rmh.nhs.uk (A.M.); shane.zaidi@rmh.nhs.uk (S.Z.); spyridon.gennatas@rmh.nhs.uk (S.G.); 2University College London Hospitals NHS Foundation Trust, London NW1 2BU, UK; katrina.ingley@nhs.net (K.I.); neha.chopra@nhs.net (N.C.); alessandra.maleddu@nhs.net (A.M.); anne.mctiernan@nhs.net (A.M.); sandra.strauss@nhs.net (S.J.S.); palma.dileo@nhs.net (P.D.); 3Sarcoma Patients EuroNet e.V./Association, D-61200 Wolfersheim, Germany; roger.wilson@sarcoma-patients.eu; 4Institute of Cancer Research, London SM2 5NG, UK; olga.husson@icr.ac.uk

**Keywords:** sarcomas, HRQoL, COVID-19, patient experience

## Abstract

Sarcomas are rare cancers with a spectrum of clinical needs and outcomes. We investigated care experiences and health-related quality of life (HRQoL) in sarcoma patients during the COVID-19 pandemic. Patients with appointments during the first two months of the UK lockdown were invited to complete a survey. Questions included views on care modifications, COVID-19 worry and psychosocial impact, and EORTC-QLQ-C30 items. 350 patients completed the survey; median age 58 (16–92) years. Care modifications included telemedicine (74%) and postponement of appointments (34%), scans (34%) or treatment (10%). Most felt the quality of care was not affected (72%), however, social life (87%) and emotional wellbeing (41%) were affected. Worry about COVID-19 infection was moderately high (mean 5.8/10) and significantly related to higher cancer-related worry; associated with lower emotional functioning irrespective of treatment intent. Curative patients (44%) with low resilient coping scores had significantly higher COVID-19 worry. Patients who did not know their treatment intent (22%) had significantly higher COVID-19 worry and insomnia. In summary, care experiences were generally positive; however, cancer-related worry, low resilient coping and uncertainty about treatment intent were associated with COVID-19 worry. These patients may benefit from additional psychological support during the pandemic and beyond.

## 1. Introduction

The current COVID-19 pandemic caused by a novel betacoronavirus (SARS-CoV-2) is a global health emergency [1,2]. As of 1 June 2020, there have been nearly 6 million confirmed cases and over 350,000 deaths across the world [2]. Mitigation and containment measures, including social distancing and quarantine, have been widely implemented in attempts to delay transmission and reduce the number of fatalities [3,4]. On 23 March 2020, the government of the United Kingdom (UK) announced a national policy mandating the entire country to stay at home except for very limited purposes such as medical need or employment in key professions [5].

Although the majority of people with COVID-19 infection will have mild–moderate symptoms (80%), the exponential rise in cases precipitated an unprecedented impact on international healthcare systems [6]. Up to 40% of hospitalised patients will require respiratory support and around 6% require mechanical ventilation [7,8,9]. Mortality rates are estimated to be between 1–4% [10]. Risk factors for death include older age, Asian and Black ethnic origin, and pre-existing medical conditions such as diabetes, hypertension, coronary artery disease and obesity [1,11]. Patients with cancer may also be more susceptible to COVID-19 and complications due to pulmonary disease, immune compromise caused by malignancy or anti-cancer treatments, and exposure to the virus through frequent visits to healthcare facilities [12]. Early data from China demonstrated that patients with cancer had a higher mortality rate, compared to patients without cancer [13]. These findings have been supported by more recent data from Europe and the United States [14,15,16]. It was therefore deemed critical to implement effective, primary preventive measures to reduce exposure and minimise risk for patients with cancer.

On 20 March 2020, the National Institute for Clinical Excellence (NICE) published guidelines for the delivery of cancer care in the UK during the COVID-19 pandemic [17]. Recommendations included postponing non-essential procedures, selective conversion of appointments to telemedicine, and prioritising patients for systemic treatment [17]. Although the impact of modifications on patients with cancer has not been described, it is widely acknowledged that the pandemic has had a negative impact on mental health and wellbeing in the general population [18,19,20,21].

Sarcomas are rare, heterogeneous cancers of mesenchymal origin that can arise in patients of any age. Prognosis is widely variable and is related to factors such as histological subtype and disease stage. Due to the wide spectrum of clinical need and outcomes for these patients, this is a unique and relevant group of patients to study during the COVID-19 pandemic, representing a microcosm of all solid cancers. The primary aims of this study were to assess the impact of the COVID-19 pandemic on care experiences, worry and health-related quality of life (HRQoL) in patients with sarcomas.

## 2. Results

### 2.1. Patient Characteristics

Three hundred and fifty patients participated in the survey (response rate: 44%; Appendix A). The median age of patients was 58 (range 16–92) years, with a slightly higher proportion of females (*n* = 192, 55%) (Table 1). The majority were of Caucasian/European ethnic origin (*n* = 286, 82%) and were well-educated (college/diploma/vocational qualification or higher) (*n* = 255, 73%). Most patients were living with others (*n* = 303, 87%) and had a partner (*n* = 245, 70%). Co-morbidities associated with poor COVID-19 outcomes were infrequent; most common was hypertension (*n* = 76, 22%). The majority of patients usually travelled more than one hour to reach their sarcoma centre (*n* = 243, 70%).

### 2.2. Loneliness

Around half the patients felt isolated; 130 patients (37%) felt isolated some of the time and 78 (11%) often felt isolated. Overall, 85 patients (24%) were lonely and 265 patients were not lonely (76%). Loneliness was associated with younger age; adolescents and young adults (aged 16–39 years; 33%), middle-aged (aged 40–64; 29%), elderly (aged ≥ 65 years; 15%); *p* = 0.004. Loneliness was also more common in patients who were not of Caucasian/European ethnic origin (36% vs. 22% Caucasian/European; *p* = 0.021), patients who were living alone (51% vs. 20% cohabiting; *p* = 0.0001), and those without a partner (41% vs. 17% with partner; *p* = 0.0001).

### 2.3. Resilient Coping

Participants had a mean resilient coping score of 14.6 (SD = 2.7) suggesting most participants were low to medium resilient copers. One hundred and fifteen patients (33%) scored less than 14 points (low resilient copers), 161 patients (46%) scored between 14 and 16 points (medium resilient copers) and 74 patients (21%) were high resilient copers.

### 2.4. Treatment Characteristics

The majority of patients reported that the intent of their treatment or management plan was curative/no evidence of disease (*n* = 150, 44%) or palliative (*n* = 117, 34%), however, 77 patients (22%) did not know their treatment intent (unknown intent). One hundred and twenty-seven patients (36%) were currently receiving treatment for sarcoma; intravenous chemotherapy (*n* = 38, 11%), oral anti-cancer treatments (*n* = 66, 19%), radiotherapy (*n* = 13, 4%) and other treatments such as microwave ablation (*n* = 10, 3%). Twenty-nine patients (8%) were taking part in a clinical trial.

### 2.5. COVID-19: Worry and Psychosocial Impact

Mean overall level of worry about the impact of COVID-19 on personal health was 5.8 out of 10 (standard deviation {SD} 2.5). Of all respondents, patients who did not know their treatment intent were the most worried about COVID-19 (6.5/10) compared to patients with palliative intent (6.0/10) or curative intent (5.4/10) treatment (*p* = 0.004). Mean level of worry about sarcoma was 5.5/10 (SD 2.6) and was highest among patients treated with palliative intent. Figure 1 demonstrates the mean level of worry about COVID-19 and sarcoma according to treatment intent and current treatment status (on active treatment versus not on treatment). In patients treated with curative intent, patients on active treatment were significantly more worried about sarcoma (*p* = 0.016) and about COVID-19 (*p* = 0.040) compared to curative patients who were not on active treatment. Patients who did not know their treatment intent (unknown group) and were on active treatment were also significantly more worried about COVID-19 (*p* = 0.005) and were more worried about sarcoma than those not on treatment, although this did not reach statistical significance (*p* = 0.054). Treatment status in palliative patients was not associated with sarcoma worry or COVID-19 worry.

Around half the patients thought that they were at a higher risk of COVID-19 than the general population (*n* = 167, 48%) (Table 2). Patients with co-morbidities associated with poor COVID-19 outcomes (56% vs. 44% no comorbidities; *p* = 0.049), who were lonely (62% vs. 38% not lonely; *p* = 0.002) or on treatment (60% vs. 40%; *p* = 0.001) were more likely to report a higher perceived risk of COVID-19. No participants had confirmed COVID-19 infection. Of the 40 patients (11%) reporting symptoms of COVID-19, six patients tested negative (2%) while others were not tested. The COVID-19 pandemic impacted finances (*n* = 84, 24%), employment (*n* = 86, 25%), emotional wellbeing (*n* = 145, 41%), family life (*n* = 210, 60%) and social life/activities (*n* = 306, 87%). The majority of patients (*n* = 255, 73%) indicated that they would accept ventilation in the event of a potentially life-threatening respiratory disease, however, a considerable proportion were not sure (*n* = 75, 22%).

### 2.6. Experiences of Care during COVID-19 Pandemic

A large proportion of patients who reported having an appointment (*n* = 286, 82%) had a telemedicine consultation (telephone/video) (*n* = 211, 74%) (Table 3). Satisfaction with telemedicine was generally high; telephone appointment mean score 8.7/10 (SD 1.7), video appointment mean score 7.4/10 (SD 3.4). Satisfaction with face to face appointments was also high 8.4/10 (SD 2.6). Nearly three-quarters of patients (*n* = 259, 74%) indicated preference for some telemedicine appointments in the future, however, 78 patients (22%) would prefer face to face appointments only. Resilient coping level (*p* = 0.024) and treatment intent (*p* = 0.047) were significantly associated with preference for face to face appointments only; patients with a low resilient coping score, and those who were uncertain about their treatment intent were most likely to prefer face to face appointments. Most participants who preferred face to face appointments indicated it was more reassuring (*n* = 58, 74%). Preference for some telemedicine appointments was due to reduced travel time (*n* = 121, *n* = 47%), cost of travel (*n* = 60, 23%), or convenience (*n* = 89, 34%). Patient characteristics (e.g., age), treatment intent and journey time to hospital were not associated with appointment preferences.

Around one third of patients had their follow-up appointments (*n* = 117, 34%) or scans (*n* = 106, 31%) postponed by at least three months (Table 3). Patient age was associated with postponement of appointments (*p* = 0.016); patients aged ≥65 years were the most likely to have postponed appointments (43%). Treatment was paused or discontinued for a minority of patients (*n* = 34, 10%). Six patients reported that trial participation had been paused due to the pandemic. A further nine patients were being considered for trial enrolment, but participation was put on hold due to the pandemic.

Postponement of treatment was significantly associated with treatment intent (*p* = 0.0001); most common among patients treated with palliative intent (Table 3). Patients generally indicated that postponement of appointments and scans were good decisions (70% and 59%, respectively). Patients treated with palliative intent were significantly more likely to be unsure whether postponement of treatment was a good or bad decision (neutral) compared to those treated with curative or unknown intent (*p* = 0.032).

Participants were most likely to contact their sarcoma clinical nurse specialist (*n* = 272, 78%) or the Royal Marsden National Health Service (NHS) Foundation Trust (RMH) or University College London Hospitals NHS Foundation Trust (UCLH) helpline (*n* = 179, 51%) in cases of cancer-related concerns. Almost two-thirds of participants felt able to contact the healthcare team as normal (*n* = 223, 64%), however, a substantial number indicated that they would only contact their team in an emergency situation (*n* = 95, 27%).

### 2.7. Health-Related Quality of Life (HRQoL)

Figure 2 compares the HRQoL of participants according to the self-reported treatment intent and the UK normative population values (non-pandemic) [22]. Significant differences according to treatment intent were observed in the domains of physical functioning (*p* = 0.0001), role functioning (*p* = 0.001), social functioning (*p* = 0.004), global health/QoL (*p* = 0.0001) and insomnia (*p* = 0.033). For these domains, patients treated with palliative intent had worse functioning, worse global HRQoL and higher insomnia scores than patients treated with curative intent. Patients who did not know the intent of their treatment had scores that fell between the curative group and palliative group for all of these domains except insomnia; these patients had the highest levels of insomnia. Cognitive and emotional functioning scores were also lower in this group, however, differences in these domains were not statistically significant. Patients treated with curative intent appeared to have higher physical, emotional and cognitive functioning, higher global HRQoL scores and lower insomnia scores, than the UK normative population during “non-pandemic” times.

### 2.8. Multivariate Analysis: Cancer Worry and COVID-19 Worry

After controlling for potential covariates, higher worry about the potential impact of COVID-19 on personal health was associated with significantly higher worry about sarcoma, irrespective of treatment intent (*p* = 0.0001) (Table 4). In patients treated with curative intent, higher COVID-19 worry was associated with lower resilient coping (*p* = 0.012). For patients who were unsure of their treatment intent (unknown intent), higher COVID-19 worry was associated with lower global HRQoL (*p* = 0.012).

Higher level of worry about sarcoma was associated with significantly lower emotional functioning across all groups. In patients treated with palliative intent, higher level of worry about sarcoma was associated with the pandemic having an impact on social life/activities (*p* = 0.039). Patients having palliative treatment and who were more worried about sarcoma were less likely to have a telemedicine appointment (*p* = 0.001). For patients who did not know their treatment intent, higher level of worry about sarcoma was associated with being on treatment (*p* = 0.038).

## 3. Discussion

To our knowledge this is the first study to assess care experiences and HRQoL among adult patients with sarcomas during the COVID-19 pandemic. A substantial proportion of follow-up appointments and scans were postponed, and many patients were monitored remotely through telemedicine (telephone or video consultations). These modifications were generally perceived to be good decisions and most patients felt able to contact their healthcare professionals as normal. Treatment was not commonly postponed; patients undergoing palliative treatment were less certain whether this was a good or bad decision. Patients with high levels of worry about cancer were the most worried about the potential impact of COVID-19 on their health and had lower emotional functioning, irrespective of treatment intent. Low resilient coping scores were also related to higher COVID-19 worry in curative patients. Early identification of patients with low resilience and cancer-related worry may facilitate timely psychological support across the disease trajectory. Patients who reported unknown treatment intent experienced high levels of worry about COVID-19 and sarcoma, lower emotional functioning, and had high levels of insomnia. Treatment goals should be discussed at appropriate time points taking into account individual preferences for information provision, such as disclosure of prognosis.

Consistent with the known risks of COVID-19 in people with cancer, half of patients perceived their own risk of COVID-19 to be greater than the general population. Patients also reported wide ranging impacts of the pandemic on their emotional wellbeing, finances, employment, family and social life, consistent with trends seen in the general population [23]. 

Consistent with national guidelines for delivery of cancer care during the COVID-19 pandemic, many patients were monitored remotely through telemedicine [17]. Face to face contact was also significantly reduced through postponement of follow-up appointments and scans where appropriate. Mehta et al. recently reported that interaction with healthcare environments was a prominent source of exposure to SARS-CoV-2 for cancer patients, emphasising the importance of avoiding non-essential visits to the hospital [14].

Treatment interruptions (postponement or cessation) were uncommon for patients in our study, however, these decisions were understandably perceived with uncertainty among some patients undergoing palliative treatment. An international survey of 149 physicians across five continents also reported reductions in all treatments including surgery, radiotherapy and chemotherapy for patients with musculoskeletal tumours [24]. Notably, palliative chemotherapy and radiotherapy were stopped or delayed in 20% and 17% of cases, respectively [24]. While the authors concluded that significant harm (pain, morbidity, loss of function) could be caused by delays to care for musculoskeletal tumours such as sarcomas, the true impact in terms of reduction in COVID-19 risk compared to cancer outcome is unknown [24]. Importantly, in our study, the vast majority of patients themselves reported the decision to delay care within the pandemic was a good decision. Follow-up studies are necessary to evaluate longer-term clinical outcomes and patient reported outcomes as it is conceivable that patient symptoms, HRQoL and prognosis may be affected.

Given the rarity and complexity of sarcomas, it is recommended that all patients are managed in a reference centre for sarcomas and/or within reference networks with multidisciplinary expertise [25,26]. However, as shown in this study, patients often live a long distance from reference centres. Rapid implementation of telemedicine during the COVID-19 pandemic has given us the opportunity to evaluate the acceptability of remote monitoring, showing that the majority of patients would prefer the incorporation of telemedicine in follow-up. Given the increasing pressures on global healthcare systems, including the National Health Service (NHS) in the UK, telemedicine may be an effective way to manage selected patients [27]. Preference for face to face appointments was associated with lower resilient coping and was more common in patients who did not know their treatment intent. These groups of patients may require more reassurance through face to face appointments. Methods to identify patient and provider preferences are needed to successfully implement telemedicine in routine care.

The prevalence of perceived social isolation in cancer patients is poorly defined, however, previous research and reports suggest it may be experienced by around a quarter of individuals [28]. Social isolation was much higher in our study, with almost half of the participants reporting that they felt isolated. The increased level of perceived isolation is likely a direct result of the COVID-19 pandemic self-isolation recommended by the UK government. A number of researchers have highlighted the negative health effects of social isolation as an unintended consequence of social distancing and the need for mitigation efforts [29,30,31,32]. This is particularly important for individuals with cancer where social isolation is associated with a number of clinical and psychosocial outcomes including mortality and HRQoL impairments [33,34]. 

In our study, 41% of patients reported that their emotional wellbeing was impacted by the COVID-19 pandemic. Interestingly, EORTC-QLQ-C30 emotional functioning scores were slightly better in patients with sarcoma (curative or palliative intent groups) compared to a normative UK population during a “non-pandemic” period [22]. Although these data are not directly comparable, adaptive coping strategies developed during cancer diagnosis and treatment may enable patients to cope with other adverse situations [35]. Patients that did not know the aim of the treatment had lower emotional functioning, higher worry about COVID-19 and higher insomnia scores. Although these findings could be attributed to uncertainty about the treatment goals, these patients may be a distinct group who require more support.

Patients who reported palliative intent of treatment/sarcoma management generally had worse HRQoL scores than those treated with curative intent. Unfortunately, EORTC-QLQ-C30 reference values for patients with sarcoma are not available for comparison and we propose repeating HRQoL analysis after the pandemic to determine whether observed trends remain and to evaluate the magnitude of those differences.

There is evidence of widespread anxiety, distress and deterioration in mental health in the general population during the COVID-19 pandemic, and a number of studies have evaluated the impact on cancer patients [36]. A Dutch study compared wellbeing in >4000 cancer patients with a matched norm-population without cancer [37]. Quality of life scores were similar; however, cancer patients were more worried about COVID-19 infection while norm-participants were significantly more likely to report depression and loneliness compared to cancer patients [37]. In another survey of breast cancer patients, around half had symptoms of anxiety, depression and insomnia, while 83% showed symptoms of distress [38]. Frey et al. evaluated the impact of COVID-19 in patients with ovarian cancer [39]. The majority reported significant cancer worry, half had borderline or abnormal anxiety, and a quarter had borderline or abnormal depression [39]. Romito et al. evaluated psychological distress in lymphoma patients and found anxiety and depressive symptoms in around one-third of patients during the pandemic [40]. Using a self-developed COVID-19 questionnaire, Falcone et al. found mean COVID-19 concern of 8/12 (interquartile range {IQR} 5–9) in thyroid cancer patients, which was inversely correlated with EORTC-QLQ-C30 emotional functioning scores [41]. Although these studies used a variety of questionnaires (validated or developed by researchers) and included patients with heterogeneous socio-demographic characteristics, different cancer types and disease stages, they further highlight the unmet need for psychological support in cancer patients during the COVID-19 pandemic and across the disease trajectory [42].

### 3.1. Implications for Care

The COVID-19 pandemic has been a period of great uncertainty. While vaccines are not available and “normality” may be many months away, we need to consider how we can identify and support patients who may be more vulnerable during the pandemic and at other stressful time points during their disease trajectory. Worry about COVID-19 was significantly associated with overall cancer-related worry, irrespective of treatment intent. Higher cancer-related worry was also associated with lower emotional functioning across all treatment intent groups. In clinical practice, screening patients for cancer-related worry may identify patients who require psychological support, both during the pandemic and beyond. Brief screening methods, such as the distress thermometer (patients rate their distress on a scale from 0 to 10), may be useful to recognise patients who may benefit from psychosocial support and enable tailored psychosocial interventions early in their cancer treatment [43]. More detailed assessments such as the Hospital and Anxiety Depression Scale (HADS) and the Brief Symptom Inventory (BSI), may also be suitable for assessing psychological concerns across the disease trajectory [44]. A number of studies have supported the efficacy of psychosocial interventions, which can be provided on an individual, family or group level. Well-established approaches for cancer patients include cognitive–behavioural, existential, and psychodynamic therapies [44]. Referral to specialist psycho-oncology services should be part of the multidisciplinary approach to cancer patient management.

We also found that lower resilient coping score was associated with higher COVID-19 worry in patients treated with curative intent. Patients who had low resilient coping scores were significantly more likely to indicate that they would prefer face to face appointments only; demonstrating that these patients may find personal interaction more reassuring. Resilience and coping are influenced by biological, personal and social factors, and there is substantial evidence that resilience is related to higher cognitive functioning and quality of life, and lower levels of anxiety [45]. We used the four-item Brief Resilient Coping Scale (BRCS) to assess resilient coping in our study. The BRCS has shown internal consistency and validity across several different diseases and populations, and it is short and easy to administer [46]. As such, this tool could be used to screen patients in clinical practice who may benefit from additional support during adverse periods; not only the COVID-19 pandemic, but other challenging clinical situations such as recurrence or progression of disease. A number of tailored interventions are available to promote resilience, improve adaptive coping strategies, alleviate distress and promote better HRQoL [47]. For example, the “Promoting Resilience in Stress Management (PRISM)” intervention, focuses on skills to manage stress through cognitive reframing, setting goals and meaning making, and has shown beneficial psychosocial outcomes in adolescents and young adults with cancer [48].

The group of patients who did not know their treatment intent (unknown intent; *n* = 77) had high worry about COVID-19 and sarcoma, insomnia and were more likely to prefer face to face appointments. Evidence indicates that treatment goals and prognosis are often not discussed due to worry that this information will take away hope or make patients depressed, or because these discussions are emotionally upsetting for clinicians themselves [49]. However, hope can be maintained through setting realistic goals, even in the setting of incurable disease [50]. Early discussions about treatment goals in the context of patient values may help to guide management decisions including timely advance care planning. Uncertainty about treatment intent in sarcoma patients requires further study.

Loneliness was associated with living alone and not having a partner. Social isolation during self-isolation when living alone is complex and these patients may be at higher risk for adverse psychosocial outcomes [29]. Loneliness was also more common among adolescents and young adults (aged 16–39 years) and middle-aged patients (aged 40–64 years). Previous studies have described loneliness and isolation among adolescents and young adults with cancer, resulting from disruption to normal developmental challenges and feeling “different” to peers [51]. Interventions that have shown promise in reducing loneliness include those focused on enhancing social skills, providing social support, facilitating opportunities for social contact and addressing maladaptive social cognition; the latter appears to be the most effective strategy [52]. The challenge is to introduce safe interventions that can support vulnerable patients and adhere to national guidelines on social distancing. Technology has been vitally important for the delivery of care during this pandemic and support could be provided through existing digital health innovations [53]. Others have tested innovative methods during the pandemic, such as a Whatsapp messaging system for resolving queries during the COVID-19 pandemic [54]. Alternative approaches must be considered for people without access to the internet. Signposting patients to accurate information will also be important, including advice about boosting wellbeing through physical activity and nutritional support.

### 3.2. Limitations

Participants in this survey were mostly White and well-educated. It would be interesting to assess the views of patients from a wider range of educational backgrounds, and also evaluate the experiences of patients from Black, Asian, and minority ethnic backgrounds; a group at particularly high risk of complications due to COVID-19. Translation of surveys and the use of other qualitative methods, such as interviews, could also be considered for future studies. Patients self-reported their treatment intent; however, previous studies have shown significant discordance between patient expectations of treatment intent [55]. As the survey was anonymous, we did not link survey responses with clinical data and therefore verification of treatment intent was not possible. The UK has a publicly-funded National Health Service (NHS), and therefore these data may not be generalizable to other countries where socioeconomic and demographic factors influence access to healthcare; such disparities may be further exaggerated during the pandemic due to variability in financial support offered by governments, public health guidance and cancer care infrastructure.

## 4. Materials and Methods

This study was a cross-sectional survey assessing the experiences of sarcoma patients at two of the largest specialist sarcoma centres in Europe, the Royal Marsden NHS Foundation Trust (RMH) and the University College London Hospitals NHS Foundation Trust (UCLH, London Sarcoma Service).

### 4.1. Participants

We identified patients aged ≥16 years with a diagnosis of sarcoma (soft tissue sarcomas, bone sarcomas and gastrointestinal stromal tumours (GIST)), who had a planned outpatient appointment between 23 March and 23 May 2020 in the Medical Oncology and Radiation Oncology departments of the Sarcoma Units at RMH and UCLH. Patients with benign tumours, including desmoid fibromatosis, that were unable to communicate in English or too unwell, were not eligible.

Eligible patients at RMH consented by phone to receive the anonymous survey (online or paper version). Eligible patients at UCLH were invited by phone or sent an email with the survey details. The survey was administered through SurveyMonkey; whose security has been established in accordance with international standards [56]. Data collection was completed 30 May 2020.

### 4.2. Measures

The survey for this study was designed by sarcoma clinicians, an epidemiology and HRQoL expert (O.H.), a patient advocate (R.W.) and the patient and public involvement panel (PPI) of the RMH (Appendix B). The survey included a combination of validated questionnaires and items developed for this study. Questions developed for this survey included socio-demographic items, sarcoma management items, worry about COVID-19, psychosocial impact of the COVID-19 pandemic, and care experiences during the pandemic. Further details and preliminary results of the telemedicine section have been published elsewhere [27].

HRQoL was assessed using the European Organization for the Research and Treatment of Cancer Quality of Life Questionnaire (EORTC-QLQ-C30) functioning scales (physical, cognitive, social, role and emotional functioning), global health status/QoL items, and the insomnia item. HRQoL were analysed using EORTC scoring guidelines, all of the scales and single-item measures range in score from 0 to 100 [57]. High scores on the scales and global HRQoL items represent high functioning or HRQoL. A high score on the insomnia item represents worse insomnia.

The UCLA (University of California, Los Angeles) abbreviated Loneliness Scale was used to assess loneliness [58]. The sum of responses to the three items of this scale indicates whether respondents are lonely (scores 6–9) or not lonely (scores 3–5). The ability to cope with stress was assessed using the Brief Resilient Coping Scale (BRCS) [46]. Summed scores correspond to “low resilient copers” (4–13 points), “medium resilient copers” (14–16) and “high resilient copers” (17–20) [45].

### 4.3. Statistical Analysis

SPSS statistics version 25 was used for the analysis. Missing responses were not included, and only available data were analysed. The online survey was programmed to ensure that all EORTC-QLQ-C30 questions were answered before responses could be submitted. Descriptive statistics were used to summarise results; frequencies and percentages for categorical items, mean and standard deviation (SD) for continuous items.

We anticipated that the COVID-19 pandemic would have a different impact on patients undergoing treatment with palliative intent compared to curative intent, and therefore stratified responses according to self-reported treatment intent (curative vs. palliative). In order to assess overall worry about sarcoma and COVID-19, we further stratified patients receiving active treatment versus those not receiving active treatment. We also included the group of patients who did not know the aim of their treatment or management as the “unknown intent” group. As EORTC-QLQ-C30 sarcoma patient reference values are not available, we included UK normative population values for comparison with HRQoL data (non-pandemic) [22].

We used Chi-squared tests to evaluate differences between categorical variables (Fisher’s exact test where cell values were ≤5). ANOVA was used to assess differences in the means between treatment groups (curative, palliative or unknown intent), including age, sarcoma worry, COVID-19 worry and HRQoL scores. We used multivariate linear regression to assess associations between patient characteristics, care modifications, psychosocial impact of pandemic, HRQoL domains, worry about COVID-19 and worry about sarcoma in each treatment group (curative/palliative/unknown intent) separately. Factors significantly associated with the independent and dependent variable in univariate analysis at the *p* ≤ 0.100 level were included in the models. *p*-values of ≤0.05 were considered statistically significant.

### 4.4. Ethical approval

Ethical approval was obtained at the RMH from the Committee for Clinical Review (SE939). Ethical approval was not required at UCLH for this survey. Anonymous survey responses from patients recruited at RMH and UCLH were combined for the final analysis. All participants gave informed consent prior to completing the survey. All data were handled in accordance with the Declaration of Helsinki.

## 5. Conclusions

Postponement of care and the use of telemedicine to replace face to face appointments were generally perceived positively, however, the longer-term clinical consequences are not yet known. Worry about potential impact of COVID-19 on personal health was moderately high and was strongly related to the level of worry about cancer and resilient coping. Emotional functioning was significantly lower in patients with high cancer-related worry, irrespective of treatment intent. Patients with high cancer-related worry or low resilience may benefit from tailored psychological support across their disease trajectory, in order to promote coping during other adverse (clinical) situations. Patients who were uncertain about their treatment intent appeared to be a distinct group with high COVID-19 worry, high insomnia and low emotional functioning. Discussions about treatment goals, which consider information preferences and beliefs, may reduce uncertainty and improve HRQoL in these patients. In view of global changes to the delivery of cancer care during the COVID-19 pandemic, our findings are of relevance to the wider cancer population. Further studies are needed to assess the longer-term impact of the pandemic on HRQoL in patients with cancer.

## Figures and Tables

**Figure 1 cancers-12-02288-f001:**
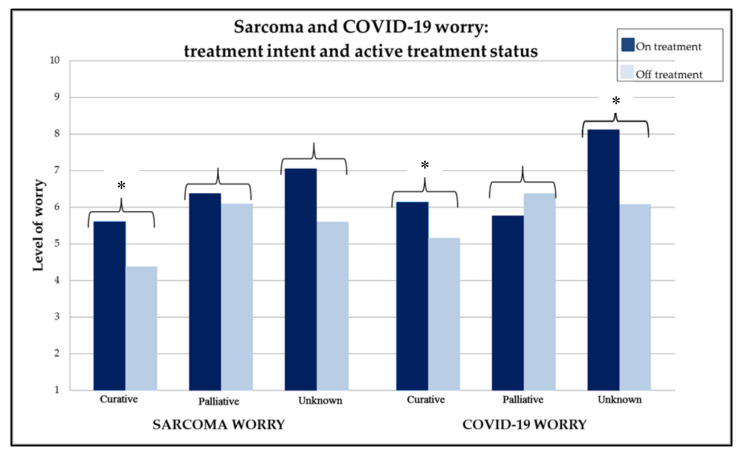
Sarcoma and COVID-19 worry according to treatment intent and active treatment status. * *p* < 0.05.

**Figure 2 cancers-12-02288-f002:**
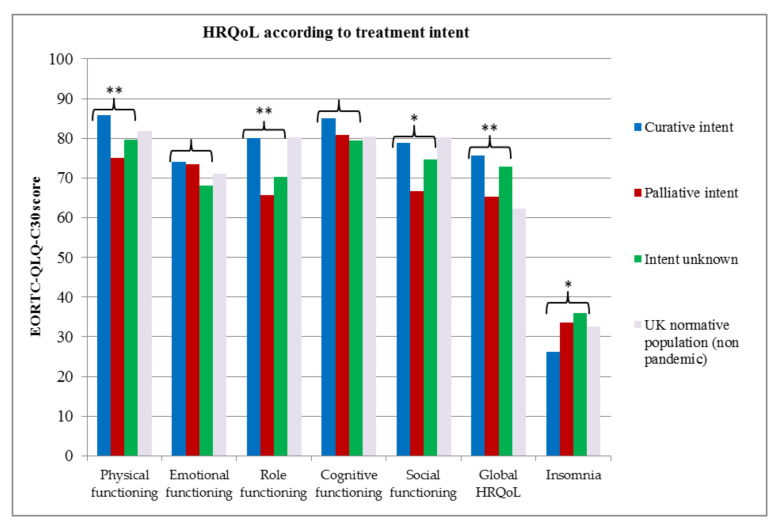
Health-related quality of life according to treatment intent. *p*-values: * *p* ≤ 0.05, ** *p* ≤ 0.001.

**Table 1 cancers-12-02288-t001:** Participant characteristics.

Patient Characteristics	All Participants (*n* = 350) *n* (%)	Curative Intent (*n* = 150) *n* (%)	Palliative Intent (*n* = 117) *n* (%)	Intent Unknown (*n* = 77) *n* (%)	*p*-Value
Age (years)					
Mean	56.1	52.0	61.4	56.3	0.060
(SD)	(17.3)	(17.9)	(14.6)	(17.8)	
Gender					
Male	158 (45)	65 (43)	58 (50)	31 (30)	0.394
Female	192 (55)	85 (57)	59 (50)	46 (60)	
Ethnicity					
Caucasian/European	286 (82)	121 (81)	100 (87)	59 (77)	0.168
Other	62 (18)	29 (19)	15 (13)	18 (23)	
Living situation					
Alone	47 (13)	16 (11)	18 (15)	12 (16)	0.439
Cohabiting	303 (87)	134 (89)	99 (85)	65 (84)	
Marital status					
Partner	245 (70)	115 (77)	81 (69)	45 (58)	0.017
No partner	105 (30)	35 (23)	36 (31)	32 (42)	
Education level					
Low	94 (27)	35 (23)	28 (24)	28 (36)	0.081
Medium	114 (33)	48 (32)	37 (32)	28 (36)	
High	141 (40)	67 (45)	51 (44)	21 (27)	
Loneliness					
Not lonely	85 (24)	120 (80)	89 (76)	52 (68)	0.115
Lonely	265 (76)	30 (20)	28 (24)	25 (32)	
Resilient coping					
Low	115 (33)	51 (34)	30 (26)	32 (42)	0.066
Medium	161 (46)	73 (49)	60 (51)	26 (34)	
High	74 (21)	26 (17)	27 (23)	19 (25)	
Comorbidities					
COPD	9 (3)	2 (1)	5 (4)	2 (3)	0.380
Hypertension	76 (22)	28 (19)	30 (26)	17 (22)	0.388
Diabetes	29 (8)	7 (5)	12 (10)	10 (13)	0.061
Coronary artery disease	17 (5)	5 (3)	10 (9)	1 (1)	0.055
Obesity	31 (9)	10 (7)	9 (8)	12 (16)	0.085
None of above	232 (66)	110 (73)	68 (58)	49 (64)	0.030
Journey to hospital > 1 h	243 (70)	93 (62)	90 (77)	57 (77)	0.014
Transport to hospital ≥ 2 modes	165 (47)	69 (46)	62 (53)	30 (39)	0.152
Cancer items					
***Current treatment***					
IV chemotherapy	38 (11)	11 (7)	24 (21)	3 (4)	0.0001
Oral chemotherapy/TKI	66 (19)	14 (9)	42 (36)	10 (13)	
Radiotherapy	13 (4)	6 (4)	6 (5)	1 (1)	
Other treatment	10 (3)	3 (2)	3 (3)	1 (1)	
No treatment	218 (63)	113 (76)	41 (35)	59 (77)	
Clinical trial					
Yes	29 (8)	6 (4)	19 (16)	4 (5)	0.001
Follow-up frequency					
Less than 3 monthly	93 (27)	25 (17)	48 (41)	19 (25)	0.0001
3–4 monthly	148 (43)	61 (41)	48 (41)	34 (44)	
6 monthly	47 (14)	29 (20)	9 (8)	9 (12)	
Annual	23 (7)	19 (13)	2 (2)	2 (3)	
Other	37 (11)	14 (10)	10 (9)	13 (17)	

**Table 2 cancers-12-02288-t002:** COVID-19 items.

COVID-19 Items	All *n* (%)	Curative Intent *n* (%)	Palliative Intent *n* (%)	Intent Unknown *n* (%)	*p*-Value
Perceived risk vs. Population					
Higher	167 (48)	52 (35)	70 (60)	44 (57)	0.0001
Equal	135 (39)	77 (51)	30 (26)	26 (34)	
Lower	30 (9)	14 (9)	13 (11)	1 (1)	
Do not know	18 (5)	7 (5)	4 (3)	6 (8)	
COVID-19 symptoms/test					
Symptoms, negative test	6 (2)	5 (3)	1 (1)	0 (0)	0.150
Symptoms, no test	34 (10)	15 (10)	13 (11)	6 (8)	
No symptoms	265 (76)	113 (75)	91 (78)	55 (71)	
Do not know	45 (13)	17 (11)	12 (10)	16 (21)	
COVID-19 pandemic impact					
Employment	86 (25)	37 (25)	32 (27)	15 (20)	0.459
Financial situation	84 (24)	33 (22)	29 (25)	21 (27)	0.661
Family life	210 (60)	85 (57)	79 (68)	42 (55)	0.112
Emotional wellbeing	145 (41)	62 (41)	48 (41)	33 (43)	0.979
Social life/activities	306 (87)	130 (87)	103 (88)	67 (87)	0.978
Would accept ventilator					
Yes	255 (73)	119 (79)	82 (70)	50 (65)	0.175
No	19 (5)	6 (4)	8 (7)	5 (7)	
Do not know	75 (22)	25 (17)	27 (23)	22 (29)	

**Table 3 cancers-12-02288-t003:** Patient reported experiences of care.

Experiences of Care	All Participants	Curative Intent	Palliative Intent	Intent Unknown	*p*-Value
Appointment type					
Face to face appointment	75 (26)	32 (27)	35 (33)	8 (14)	0.003
Telemedicine appointment	211 (74)	86 (73)	70 (67)	51 (86)	
Preferred appointment type					
Face to face (F2F) only *	78 (22)	29 (19)	25 (22)	23 (30)	0.047
Mostly F2F, occasional telemedicine	129 (37)	68 (45)	36 (31)	23 (30)	
Mostly telemedicine, occasional F2F	118 (34)	44 (29)	48 (41)	23 (30)	
Only telemedicine	12 (4)	6 (4)	4 (3)	2 (3)	
Unsure	12 (4)	3 (2)	3 (3)	6 (8)	
Appointment modification					
Postponed	117 (34)	48 (33)	40 (35)	28 (36)	0.853
Opinion on postponement					
Positive	82 (70)	39 (81)	24 (60)	18 (64)	0.220
Negative	10 (9)	2 (4)	4 (10)	4 (14)	
Neutral	24 (21)	7 (15)	11 (28)	6 (21)	
Imaging modification					
Postponed	106 (31)	42 (29)	33 (30)	28 (37)	0.422
Opinion on postponement					
Positive	63 (59)	28 (67)	18 (55)	14 (50)	0.212
Negative	11 (10)	2 (5)	3 (9)	6 (21)	
Neutral	32 (30)	12 (29)	12 (36)	8 (29)	
Treatment modification					
Postponed	29 (9)	5 (3)	20 (18)	4 (5)	0.0001
Discontinued	5 (2)	2 (1)	3 (3)	0 (0)	
Opinion on postponed/stopped					
Positive	14 (48)	3 (75)	8 (40)	3 (75)	0.093
Negative	2 (7)	1 (25)	1 (5)	0 (0)	
Neutral	12 (41)	0 (0)	11 (55)	1 (25)	
Impact of pandemic on care quality					
Positive	4 (1)	3 (2)	0 (0)	1 (1)	0.155
Negative	50 (15)	18 (12)	23 (20)	8 (10)	
Not affected	250 (72)	113 (76)	78 (68)	55 (71)	
Unsure	42 (12)	14 (10)	14 (12)	13 (17)	
Informed about care plan					
Very well informed	190 (55)	94 (63)	57 (49)	35 (47)	0.017
Informed	126 (36)	44 (30)	53 (45)	27 (36)	
Little information	23 (7)	8 (5)	5 (4)	10 (13)	
Not informed	8 (2)	3 (2)	2 (2)	3 (4)	
Contacting HCP during pandemic **					
Contact as normal	223 (64)	98 (66)	79 (68)	44 (57)	0.019
Worried about availability	19 (5)	6 (4)	10 (9)	2 (3)	
Only if essential	95 (27)	39 (26)	28 (24)	25 (33)	
Do not know	12 (3)	6 (4)	0 (0)	6 (8)	
Contact for cancer support					
Clinical nurse specialist	272 (78)	114 (76)	96 (82)	60 (78)	0.482
Sarcoma helpline (RMH/UCLH)	179 (51)	80 (53)	61 (52)	33 (43)	0.300
General practitioner (GP)	72 (21)	29 (19)	22 (19)	29 (26)	0.420
NHS helpline	15 (4)	5 (3)	4 (3)	5 (7)	0.474
Sarcoma charity	19 (5)	10 (7)	5 (4)	4 (5)	0.690
Cancer charity (any)	44 (13)	14 (9)	22 (19)	8 (10)	0.055
Online peer support	16 (5)	6 (4)	6 (5)	4 (5)	0.880

Abbreviations: * F2F: face to face. ** HCP: healthcare professional.

**Table 4 cancers-12-02288-t004:** Multivariate linear regression: factors associated with sarcoma worry and COVID-19 worry.

	Worry about COVID-19 on Health			Worry about Sarcoma		
**Patient Factors**	**Curative Intent**	**Palliative Intent**	**Unknown Intent**	**Curative Intent**	**Palliative Intent**	**Unknown Intent**
Age (continuous)	-	-	-	-	*β* = −0.026 *p* = 0.782	-
Ethnicity (Caucasian vs. other)	-	-	*β* = 0.102 *p* = 0.233	-	-	-
Loneliness (continuous)	*β* = 0.077 *p* = 0.393	-	-	*β* = −0.110 *p* = 0.242	-	-
Resilient coping (continuous)	*β* = −0.206 *p* = 0.012	-	-	*β* = −0.060 *p* = 0.475	-	-
Comorbidities (Yes/No)	-	-	-	-	*β* = −0.127 *p* = 0.160	-
**Care Factors**						
On treatment (Yes/No)	*β* = 0.056 *p* = 0.475	-	*β* = 0.175 *p* = 0.062	*β* = 0.116 *p* = 0.183	-	*β* = 0.221 *p* = 0.038
Telemedicine (Yes/No)	-	-	-	*β* = −0.101 *p* = 0.232	*β* = −0.282 *p* = 0.001	*β* = −0.069 *p* = 0.525
Appointment postponed (Yes/No)	-	-	-	*β* = −0.057 *p* = 0.494	-	-
Treatment postponed (Yes/No)	-	-	-	-	*β* = 0.094 *p* = 0.260	-
**Impact of Pandemic**						
Financial impact (Yes/No)	-	-	*β* = 0.037 *p* = 0.690	-	*β* = 0.064 *p* = 0.438	-
Family impact (Yes/No)	-	*β* = 0.090 *p* = 0.300	*β* = 0.036 *p* = 0.696	-		*β* = 0.041 *p* = 0.716
Emotional impact (Yes/No)	*β* = 0.173 *p* = 0.056	*β* = 0.175 *p* = 0.071	*β* = 0.048 *p* = 0.620	-	*β* = −0.060 *p* = 0.541	*β* = 0.180 *p* = 0.117
Social life impact (Yes/No)	-	*β* = 0.119 *p* = 0.173		-	*β* = 0.171 *p* = 0.039	-
**HRQoL**						
Sarcoma worry (continuous)	*β* = 0.340 *p* = 0.0001	*β* = 0.464 *p* = 0.0001	*β* = 0.582 *p* = 0.0001	-	-	-
Physical functioning (continuous)	*β* = −0.187 *p* = 0.122	*β* = −0.011 *p* = 0.928	*β* = −0.009 *p* = 0.941	*β* = −0.051 *p* = 0.686	*β* = −0.037 *p* = 0.758	*β* = 0.114 *p* = 0.424
Emotional functioning (continuous)	*β* = 0.054 *p* = 0.661	*β* = 0.011 *p* = 0.923	*β* = 0.105 *p* = 0.427	*β* = −0.409 *p* = 0.0001	*β* = −0.410 *p* = 0.0001	*β* = −0.572 *p* = 0.0001
Role functioning (continuous)	*β* = 0.065 *p* = 0.547	*β* = −0.058 *p* = 0.644	*β* = 0.013 *p* = 0.915	*β* = −0.103 *p* = 0.370	*β* = 0.101 *p* = 0.423	*β* = −0.101 *p* = 0.510
Cognitive functioning (continuous)	*β* = 0.030 *p* = 0.778	*β* = −0.109 *p* = 0.283	*β* = 0.136 *p* = 0.218	*β* = 0.215 *p* = 0.061	-	*β* = 0.046 *p* = 0.735
Social functioning (continuous)	*β* = −0.129 *p* = 0.218	*β* = 0.012 *p* = 0.925	*β* = 0.067 *p* = 0.571	*β* = −0.098 *p* = 0.364	*β* = −0.201 *p* = 0.127	*β* = −0.187 *p* = 0.192
Global HRQoL (continuous)	*β* = 0.064 *p* = 0.540	*β* = 0.108 *p* = 0.347	*β* = −0.285 *p* = 0.012	*β* = 0.032 *p* = 0.775	*β* = −0.051 *p* = 0.662	*β* = 0.117 *p* = 0.404
Insomnia (continuous)	*β* = −0.042 *p* = 0.639	-	*β* = 0.238 *p* = 0.053	*β* = 0.088 *p* = 0.353	-	*β* = −0.166 *p* = 0.259

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
