# Peer review of "Health-Related Quality of Life and Experiences of Sarcoma Patients during the COVID-19 Pandemic"

_cancers, 2020, doi:10.3390/cancers12082288_

Round 1

Reviewer 1 Report

The article is original and I read it with interest as an orthopaedic surgeon with a special interest in orthopaediconcology.

Author Response

Reviewer 1

Comment 1:

The article is original and I read it with interest as an orthopaedic surgeon with a special interest in orthopaediconcology.

Response:

We would like to thank the reviewer for their positive feedback.

Reviewer 2 Report

Manuscript entitled "Health-related quality of life and experiences of sarcoma patients during the COVID-19 pandemic "

I am sorry to say that this work is not scientifically sound. It is of vary limited value in cancer area. It belongs "healthcare" area. I suggest this work should be transferred and not be published here. 

Author Response

Reviewer 2

Comment 1:

Manuscript entitled "Health-related quality of life and experiences of sarcoma patients during the COVID-19 pandemic." I am sorry to say that this work is not scientifically sound. It is of vary limited value in cancer area. It belongs "healthcare" area. I suggest this work should be transferred and not be published here. 

Response:

Patient reported outcomes, such as health-related quality of life (HRQoL), are recognized by the global oncology community as a key component of patient-centered care. In the current study we used several validated questionnaires, including the EORTC-QLQ-C30, which has been used to assess HRQoL in several thousand peer-reviewed cancer-related publications. Although COVID-19-specific questionnaires were not available at the time of writing, COVID-19 items were developed in collaboration with a patient advocate and quality of life expert. These enabled a unique insight into the impact of the COVID-19 pandemic on psychosocial wellbeing during this period of uncertainty.

Patients with sarcomas often have a high burden of symptoms and poor prognosis compared to many other cancers. Despite this, there is a paucity of literature in sarcoma patients related to wellbeing and HRQoL. Our study of 350 sarcoma patients is therefore relevant to sarcoma patients and healthcare professionals across the world. For example, high acceptability of telemedicine among almost all patients may be practice changing for patients with sarcomas who often live far away from expert centers. In view of global modifications to cancer care, our findings will also serve as baseline data to determine whether changes to treatment and follow-up have had an impact on HRQoL.  Furthermore, we identified that cancer-related worry was strongly related to poor emotional functioning, highlighting the unmet need for psychological support among cancer patients. We therefore feel that our work is highly relevant to cancer researchers with an interest in quality of life, and more specifically to sarcoma patients and experts. Cancers has previously published studies on cancer survivorship outcomes including quality of life.

Reviewer 3 Report

The authors present a cross-sectional evaluation of health-related quality of life and the impact of changes in care experiences due to the ongoing COVID-19 pandemic at 2 sarcoma programs in the United Kingdom. Overall the authors find that many patients did not report a change in the quality of their care with postponement of appointments or telemedicine. Certain groups of patients were identified as having higher risk of worry about COVID-19 such as those with low resilient coping skills, and patients who did not have clear knowledge of their treatment intent.

The manuscript is well-written and the methodology is sound. The authors may wish to consider separately evaluating patients actively receiving therapy vs. those not currently receiving treatment, particularly among the population of patients who received curative intent (for curative/no treatment, n=113, ~1/3 of the total study group). Concerns about health in relation to sarcoma treatment may be quite different for subjects who are off therapy and consider themselves "cured" or "in remission" than those on active treatment. A limitation that should additionally be considered in the generalizability of these results is the access to nationally funded healthcare via the NHS; disparities in healthcare access along with other socioeconomic and demographic factors may have a significant impact in countries such as the United States.

Minor comment: Please denote the meaning of * and ** in the footer to Figure 2.

Author Response

Reviewer 3

Reviewer 4 Report

It’s an interesting article to discuss “Health-related quality of life and experiences of sarcoma patients during the COVID-19 pandemic”. And, the results (telemedicine, HRQoL, isolation, loneliness) are also meaningful for other sarcoma patients and hospitals. We are looking forward to your next study with “Follow-up studies are necessary to evaluate longer-term clinical outcomes and patient reported outcomes as it is conceivable that patient symptoms, HRQoL and prognosis may be affected.”

Author Response

Reviewer 4

Comment 1

It’s an interesting article to discuss “Health-related quality of life and experiences of sarcoma patients during the COVID-19 pandemic”. And, the results (telemedicine, HRQoL, isolation, loneliness) are also meaningful for other sarcoma patients and hospitals. We are looking forward to your next study with “Follow-up studies are necessary to evaluate longer-term clinical outcomes and patient reported outcomes as it is conceivable that patient symptoms, HRQoL and prognosis may be affected.

Response 1

We would like to thank the reviewer for their positive feedback. We agree that these data will serve as useful baseline data in order to evaluate the longer-term impact of the pandemic and care modifications on outcomes including HRQoL.